# The streaming rollout of deep networks - towards fully model-parallel execution

**Volker Fischer**
Bosch Center for Artificial Intelligence
Renningen, Germany
volker.fischer@de.bosch.com

**Jan Köhler**
Bosch Center for Artificial Intelligence
Renningen, Germany
jan.koehler@de.bosch.com

**Thomas Pfeil**
Bosch Center for Artificial Intelligence
Renningen, Germany
thomas.pfeil@de.bosch.com

## Abstract

Deep neural networks, and in particular recurrent networks, are promising candidates to control autonomous agents that interact in real-time with the physical world. However, this requires a seamless integration of temporal features into the network's architecture. For the training of and inference with recurrent neural networks, they are usually rolled out over time, and different rollouts exist. Conventionally during inference, the layers of a network are computed in a sequential manner resulting in sparse temporal integration of information and long response times. In this study, we present a theoretical framework to describe rollouts, the level of model-parallelization they induce, and demonstrate differences in solving specific tasks. We prove that certain rollouts, also for networks with only skip and no recurrent connections, enable earlier and more frequent responses, and show empirically that these early responses have better performance. The *streaming* rollout maximizes these properties and enables a fully parallel execution of the network reducing runtime on massively parallel devices. Finally, we provide an open-source toolbox to design, train, evaluate, and interact with streaming rollouts.

## 1 Introduction

Over the last years, the combination of newly available large datasets, parallel computing power, and new techniques to implement and train deep neural networks has led to significant improvements in the fields of vision [1], speech [2], and reinforcement learning [3]. In the context of autonomous tasks, neural networks usually interact with the physical world in real-time which renders it essential to integrate the processing of temporal information into the network's design.

Recurrent neural networks (RNNs) are one common approach to leverage temporal context and have gained increasing interest not only for speech [4] but also for vision tasks [5]. RNNs use neural activations to inform future computations, hence introducing a recursive dependency between neuron activations. This augments the network with a memory mechanism and allows it, unlike feed-forward neural networks, to exhibit dynamic behavior integrating a stream or sequence of inputs. For training and inference, backpropagation through time (BPTT) [6] or its truncated version [6, 7] are used, where the RNN is *rolled out* (or unrolled) through time disentangling the recursive dependencies and transforming the recurrent network into a feed-forward network.

Since unrolling a cyclic graph is not well-defined [8], different possible rollouts exist for the same neural network. This is due to the rollout process itself, as there are several ways to unroll cycles with length greater 1 (larger cycles than recurrent self-connections). More general, there are two ways to unroll every edge (cf. Fig. 1): having the edge connect its source and target nodes at the same point in time (see, e.g., vertical edges in Fig. 1b) or bridging time steps (see, e.g., Fig. 1c). Bridging is especially necessary for self-recurrent edges or larger cycles in the network, so that the rollout in fact becomes a feed-forward network. In a rollout, conventionally most edges are applied in the intra-frame non-bridging manner and bridge time steps only if necessary [9, 10, 11, 12]. We refer to these rollouts as *sequential* rollouts throughout this work. One contribution of this study is the proof that the number of rollouts increases exponentially with network complexity.

The main focus of this work is that different rollouts induce different levels of model-parallelism and different behaviors for an unrolled network. In rollouts inducing complete model-parallelism, which we call *streaming*, nodes of a certain time step in the unrolled network become computationally disentangled and can be computed in parallel (see Fig. 1c). This idea is not restricted to recurrent networks, but generalizes to a large variety of network architectures covered by the presented graph-theoretical framework in Sec. 3. In Sec. 4, we show experimental results that emphasize the difference of rollouts for both, networks with recurrent and skip, and only skip connections. In this study, we are not concerned comparing performances between networks, but between different rollouts of a given network (e.g., Fig. 1b vs. c).

Our theoretical and empirical findings show that streaming rollouts enable fully model-parallel inference achieving low-latency and high-frequency responses. These features are particularly important for real-time applications such as autonomous cars [13] or UAV systems [14] in which the neural networks have to make complex decisions on high dimensional and frequent input signals within a short time.

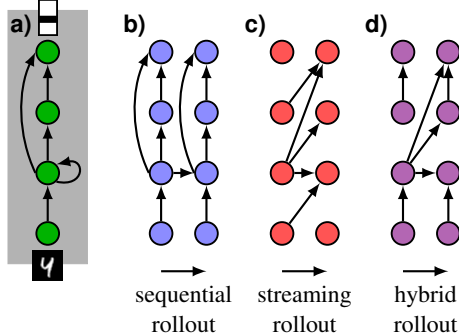

Figure 1: (best viewed in color) **a:** Neural network with skip and recurrent connections (SR) and different rollouts: **b:** the sequential rollout, **c:** the streaming rollout and **d:** a hybrid rollout. Nodes represent layers, edges represent transformations, e.g., convolutions. Only one rollout step is shown and each column in (b-d) is one frame within the rollout.

To the best of our knowledge, up to this study, no general theory exists that compares different rollouts and our contributions can be summarized as follows:

- We provide a theoretical framework to describe rollouts of deep neural networks and show that, and in some cases how, different rollouts lead to different levels of model-parallelism and network behavior.

- We formally introduce streaming rollouts enabling fully model-parallel network execution, and mathematically prove that streaming rollouts have the shortest response time to and highest sampling frequency of inputs.

- We empirically give examples underlining the theoretical statements and show that streaming rollouts can further outperform other rollouts by yielding better early and late performance.

- We provide an open-source toolbox specifically designed to study streaming rollouts of deep neural networks.

## 2   Related work

The idea of RNNs dates back to the mid-70s [15] and was popularized by [16]. RNNs and their variants, especially Long Short-Term Memory networks (LSTM) [17], considerably improved performance in different domains such as speech recognition [4], handwriting recognition [5], machine translation [18], optical character recognition (OCR) [19], text-to-speech synthesis [20], social signal classification [21], or online multi-target tracking [22]. The review [23] gives an overview of the history and benchmark records set by DNNs and RNNs.

**Variants of RNNs:** There are several variants of RNN architectures using different mechanisms to memorize and integrate temporal information. These include LSTM networks [17] and related architectures like Gated Recurrent Unit (GRU) networks [24] or recurrent highway networks [25]. Neural Turing Machines (NTM) [26] and Differentiable Neural Computers (DNC) [27] extend RNNs by an addressable external memory. Bi-directional RNNs (BRNNs) [28] incorporate the ability to model the dependency on future information. Numerous works extend and improve these RNN variants creating architectures with advantages for training or certain data domains (e.g., [29, 30, 31, 32]).

**Response time:** While RNNs are the main reason to use network rollouts, in this work we also investigate rollouts for non-recurrent networks. Theoretical and experimental results suggest that different rollout types yield different behavior especially for networks containing skip connections. The rollout pattern influences the *response time* of a network which is the duration between input (stimulus) onset and network output (response).

Shortcut or skip connections can play an important role to decrease response times. Shortcut branches attached to intermediate layers allow earlier predictions (e.g., BranchyNet [33]) and iterative predictions refine from early and coarse to late and fine class predictions (e.g., feedback networks [12]). In [34], the authors show that identity skip connections, as used in Residual Networks (ResNet) [1], can be interpreted as local network rollouts acting as filters, which could also be achieved through recurrent self-connections. The good performance of ResNets underlines the importance of local recurrent filters. The runtime of inference and training for the same network can also be reduced by network compression [35, 36] or optimization of computational implementations [37, 38].

**Rollouts:** To train RNNs, different rollouts are applied in the literature, though lacking a theoretically founded background. One of the first to describe the transformation of a recurrent MLP into an equivalent feed-forward network and depicting it in a streaming rollout fashion was [39, ch. 9.4]. The most common way in literature to unroll networks over time is to duplicate the model for each time step as depicted in Fig. 1b [ch. 10.1 in 40, 9, 10, 11, 12, 41]. However, as we will show in this work, this rollout pattern is neither the only way to unroll a network nor the most efficient.

The recent work of Carreira et al. [42] also addresses the idea of model-parallelization through dedicated network rollouts to reduce latency between input and network output by distributing computations over multiple GPUs. While their work shows promising empirical findings in the field of video processing, our work provides a theoretical formulation for a more general class of networks and their rollouts. Our work also differs in the way the delay between input and output, and network training is addressed.

Besides the chosen rollout, other methods exist, that modify the integration of temporal information: for example, *temporal stacking* (convolution over time), which imposes a fixed temporal receptive field (e.g., [43, 44]), *clocks*, where different parts of the network have different update frequencies, (e.g., [45, 46, 47, 48, 49]) or *predictive states*, which try to compensate temporal delays between different network parts (e.g., [42]). For more details, please see also Sec. 5.

## 3 Graph representations of network rollouts

We describe dependencies inside a neural network $N$ as a directed graph $N = (V, E)$. The nodes $v \in V$ represent different layers and the edges $e \in E \subset V \times V$ represent transformations introducing direct dependencies between layers. We allow self-connections $(v, v) \in E$ and larger cycles in a network. Before stating the central definitions and propositions, we introduce notations used throughout this section and for the proofs in the appendix.

Let $G = (V, E)$ be a directed graph with vertices (or nodes) $v \in V$ and edges $e = (e_{\text{src}}, e_{\text{tgt}}) \in E \subset V \times V$. Since neural networks process input data, we denote the **input** of the graph as set $I_G$, consisting of all nodes without incoming edges:

$$I_G := \{v \in V \mid \nexists u \in V : (u, v) \in E\}. \tag{1}$$

A **path** in $G$ is a mapping $p : \{1, \ldots, L\} \to E$ with $p(i)_{\text{tgt}} = p(i+1)_{\text{src}}$ for $i \in \{1, \ldots, L-1\}$ where $L \in \mathbb{N}$ is the **length** of $p$. We denote the length of a path $p$ also as $|p|$ and the number of elements in a set $A$ as $|A|$. A path $p$ is called **loop** or **cycle** iff $p(|p|)_{\text{tgt}} = p(1)_{\text{src}}$ and it is called **minimal** iff $p$ is injective. The set of all cycles is denoted as $C_G$. Two paths are called **non-overlapping** iff they share

no edges. We say a graph is **input-connected** iff for every node $v$ exists a path $p$ with $p(|p|)_{\text{tgt}} = v$ and $p(1)_{\text{src}} \in I_G$. Now we proceed with our definition of a (neural) network.

**Definition (network):** A **network** is a directed and input-connected graph $N = (V, E)$ for which $0 < |E| < \infty$.

For our claims, this abstract formulation is sufficient and, while excluding certain artificial cases, it ensures that a huge variety of neural network types is covered (see Fig. A1 for network examples). For deep neural networks, we give an explicit formulation of this abstraction in Sec. A1.2, which we also use for our experiments. Important concepts introduced here are illustrated in Fig. 2. In this work, we separate the concept of network rollouts into two parts: The temporal propagation scheme which we call *rollout pattern* and its associated *rollout windows* (see also Fig. 1 and Fig. 2):

**Definition (rollout pattern and window):** Let $N = (V, E)$ be a network. We call a mapping $R : E \rightarrow \{0, 1\}$ a **rollout pattern** of $N$. For a rollout pattern $R$, the **rollout window** of size $W \in \mathbb{N}$ is the directed graph $R_W = (V_W, E_W)$ with:

$$
\begin{aligned}
V_W &:= \{0, \ldots, W\} \times V, \quad \overline{v} = (i, v) \in V_W \\
E_W &:= \{((i, u), (j, v)) \in V_W \times V_W \mid (u, v) \in E \;\wedge\; j = i + R((u, v))\}.
\end{aligned}
\tag{2}
$$

Edges $e \in E$ with $R(e) = 1$ enable information to directly *stream* through time. In contrast, edges with $R(e) = 0$ cause information to be processed within frames, thus introducing *sequential* dependencies upon nodes inside a frame. We dropped the dependency of $E_W$ on the rollout pattern $R$ in the notation. A rollout pattern and its rollout windows are called **valid** iff $R_W$ is acyclic for one and hence for all $W \in \mathbb{N}$. We denote the set of all valid rollout patterns as $\mathcal{R}_N$ and the rollout pattern $R \equiv 1$ the **streaming rollout** $R^{\text{stream}} \in \mathcal{R}_N$. We say two rollout patterns $R$ and $R'$ are **equally model-parallel** iff they are equal ($R(e) = R'(e)$) for all edges $e = (u, v) \in E$, not originating in the network's input ($u \notin I_N$). For $i \in \{0, \ldots, W\}$, the subset $\{i\} \times V \subset V_W$ is called the $i$-th **frame**.

**Proof:** In Sec. A1.3, we prove that the definition of valid rollout patterns is well-defined and is consistent with intuitions about rollouts, such as consistency over time. We also prove that the streaming rollout exists for every network and is always valid.

The most non-streaming rollout pattern $R \equiv 0$ is not necessarily valid, because if $N$ contains loops then $R \equiv 0$ does not yield acyclic rollout windows. Commonly, recurrent networks are unrolled such that most edges operate inside the same frame ($R(e) = 0$), and only when necessary (e.g., for recurrent or top-down) connections are unrolled ($R(e) = 1$). In contrast to this sequential rollout, the streaming rollout pattern unrolls all edges with $R(e) = 1$ (cf. top and third row in Fig. 2).

**Lemma 1:** Let $N = (V, E)$ be a network. The number of valid rollout patterns $|\mathcal{R}_N|$ is bounded by:

$$
1 \leq n \leq |\mathcal{R}_N| \leq 2^{|E| - |E_{\text{rec}}|},
\tag{3}
$$

where $E_{\text{rec}}$ is the set of all self-connecting edges $E_{\text{rec}} := \{(u, v) \in E \mid u = v\}$, and $n$ either:

- $n = 2^{|E_{\text{forward}}|}$, with $E_{\text{forward}}$ being the set of edges not contained in any cycle of $N$, or
- $n = \prod\limits_{p \in C} (2^{|p|} - 1)$, $C \subset C_N$ being any set of minimal and pair-wise non-overlapping cycles.

**Proof:** See appendix Sec. A1.4.

Lemma 1 shows that the number of valid rollout patterns increases exponentially with network complexity. Inference of a rollout window is conducted in a sequential manner. This means, the state of all nodes in the rollout window is successively computed depending on the availability of already computed source nodes[1]. The chosen rollout pattern determines the mathematical function this rollout represents, which may be different between rollouts, e.g., for skip connections. In addition, the chosen rollout pattern also determines the order in which nodes can be computed leading to different runtimes to compute the full state of a rollout window.

We now introduce tools to compare these addressed differences between rollouts. *States* of the rollout window encode, which nodes have been computed so far and *update steps* determine the next state

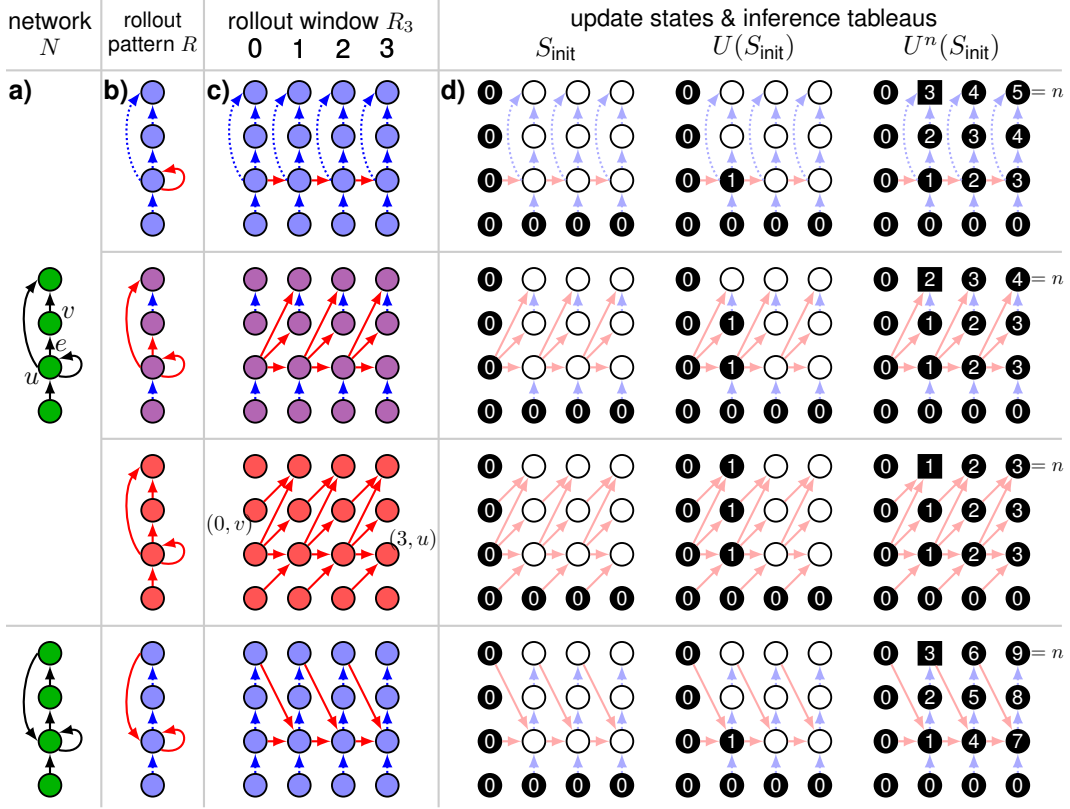

Figure 2: (best viewed in color) **a:** Two different networks. **b:** Different rollout patterns $R : E \to \{0, 1\}$ for the two networks. *Sequential* ($R(e) = 0$) and *streaming* ($R(e) = 1$) edges are indicated with blue dotted and red solid arrows respectively. For the first network (top to bottom), the most sequential, one hybrid, and the streaming rollout patterns are shown. For the second network, one out of its 3 most sequential rollout patterns is shown (either of the three edges of the cycle could be unrolled). **c:** Rollout windows of size $W = 3$. By definition (Eq. (2)), sequential and streaming edges propagate information within and to the next frames respectively. **d:** States $S(\overline{v})$ and inference tableau values $T(\overline{v})$. The state $S(\overline{v})$ of a node is indicated with black (already known) or white (not yet computed). From left to right: initial state $S_{\text{init}}$, state after first update step $U(S_{\text{init}})$, full state $U^n(S_{\text{init}}) = S_{\text{full}}$. The number of update steps $n$ to reach the full state differs between rollouts. Numbers inside nodes $\overline{v}$ indicate values of the inference tableau ($T(\overline{v})$). Inference factors $F(R)$ are indicated with square instead of circular nodes in the first frame of the full states.

based on the previous state. *Update tableaus* list after how many update steps nodes in the rollout window are computed. Update states, update steps, and inference tableaus are shown for example networks and rollouts in Fig. 2.

**Definition (update state, update step, tableau, and factor):**   Let $R$ be a valid rollout pattern of a network $N = (V, E)$. A **state** of the rollout window $R_W$ is any mapping $S : V_W \to \{0, 1\}$. Let $\Sigma_W$ denote the set of all possible states. We define the **full state** $S_{\text{full}}$ and **initial state** $S_{\text{init}}$ as:

$$S_{\text{full}} \equiv 1; \qquad S_{\text{init}}((i, v)) = 1 \iff v \in I_N \vee i = 0. \qquad (4)$$

Further, we define the **update step** $U$ which updates states $S$. Because the updated state $U(S)$ is again a state and hence a mapping, we define $U$ by specifying the mapping $U(S)$:

$$U : \Sigma_W \to \Sigma_W; \qquad U(S) : V_W \to \{0, 1\} \qquad (5)$$

$$U(S)(\overline{v}) := \begin{cases} 1 & \text{if } S(\overline{v}) = 1 \text{ or if for all } (\overline{u}, \overline{v}) \in E_W : S(\overline{u}) = 1 \\ 0 & \text{otherwise} \end{cases}$$

We call the mapping $T : V_W \to \mathbb{N}$ the **inference tableau**:

$$T(\overline{v}) \;:=\; \max_{p \in P_{\overline{v}}} |p| \;=\; \operatorname*{argmin}_{n \in \mathbb{N}} \{ U^n(S_{\text{init}})(\overline{v}) = 1 \} \tag{6}$$

where $U^n$ is the $n$-th recursive application of $U$ and for $\overline{v} \in V_W$, $P_{\overline{v}}$ denotes the set of all paths in $R_W$ that end at $\overline{v}$ (i.e., $p(|p|)_{\text{tgt}} = \overline{v}$) and for which their first edge may start but not end in the 0-th frame, $p(1)_{\text{tgt}} \notin \{0\} \times V$. Hereby, we exclude edges (computational dependencies) which never have to be computed, because all nodes in the 0-th frame are initialized from start. We dropped the dependencies of $U$ and $T$ on the rollout window $R_W$ in the notation and if needed we will express them with $U_{R_W}$ and $T_{R_W}$. Further, we call the maximal value of $T$ over the rollout window of size 1 the rollout pattern's **inference factor**:

$$F(R) := \max_{\overline{v} \in V_1} T_{R_1}(\overline{v}). \tag{7}$$

**Proof:** In Sec. A1.6 we prove Eq. (6).

We also want to note that all rollout windows of a certain window size $W$ have the same number of edges $W * |E|$, independent of the chosen rollout pattern (ignoring edges inside the 0-th frame, because these are not used for updates). However, maximal path lengths in the rollout windows differ between different rollout patterns (cf. Eq. (6) and its proof, as well as tableau values in Fig. 2).

Inference of rollout windows starts with the initial state $S_{\text{init}}$. Successive applications of the update step $U$ updates all nodes until the fully updated state $S_{\text{full}}$ is reached (cf. Fig. 2 and see Sec. A1.5 for a proof). For a certain window size $W$, the number of operations to compute the full state is independent of the rollout pattern, but which updates can be done in parallel heavily depends on the chosen rollout pattern. We will use the number of required update steps to measure computation time. This number differs between different rollout patterns (e.g., $F(R)$ in Fig. 2). In practice, the time needed for the update $U(S)$ of a certain state $S$ depends on $S$ (i.e., which nodes can be updated next). For now, we will assume independence, but will address this issue in the discussion (Sec. 5).

**Theorem 1:** Let $R$ be a valid rollout pattern for a network $N = (V, E)$ then the following statements are equivalent:

   a) $R$ and the streaming rollout pattern $R^{\text{stream}}$ are equally model-parallel.

   b) The first frame is updated entirely after the first update step: $F(R) = 1$.

   c) For $W \in \mathbb{N}$, the $i$-th frame of $R_W$ is updated at the $i$-th update step:

$$\forall (i, v) \in V_W \;:\; T((i, v)) \le i.$$

   d) For $W \in \mathbb{N}$, the inference tableau of $R_W$ is minimal everywhere and over all rollout patterns. In other words, responses are earliest and most frequent:

$$\forall \overline{v} \in V_W \;:\; T_{R_W}(\overline{v}) = \min_{R' \in \mathcal{R}_N} T_{R'_W}(\overline{v}).$$

**Proof:** See appendix Sec. A1.7.

## 4   Experiments

To demonstrate the significance of the chosen rollouts w.r.t. the runtime for inference and achieved accuracy, we compare the two extreme rollouts: the most model-parallel, i.e., streaming rollout ($R \equiv 1$, results in red in Fig. 3), and the most sequential rollout[2] ($R(e) = 0$ for maximal number of edges, results in blue in Fig. 3).

In all experiments, we consider a response time task, in which the input is a sequence of images and the networks have to respond as quickly as possible with the correct class. We want to restate that we do not compare performances between networks but between rollout patterns of the same network.

For all experiments and rollout patterns under consideration, we conduct inference on shallow rollouts ($W = 1$) and initialize the zero-th frame of the next rollout window with the last (i.e., 1.) frame of

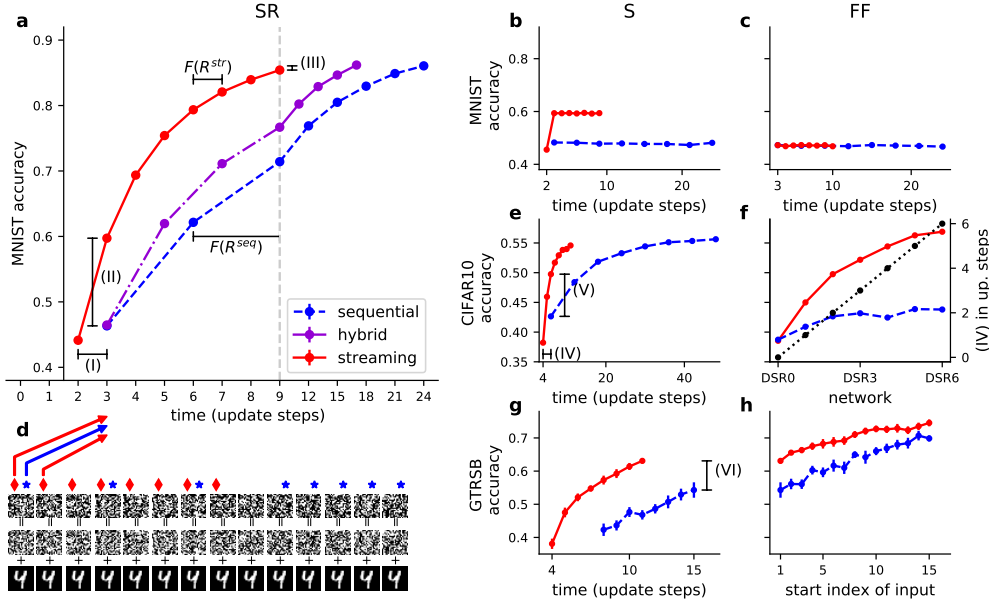

Figure 3: (best viewed in color) Classification accuracy for sequential (in dashed blue), streaming (in solid red), and one hybrid (violet; only for SR network in a) rollout on MNIST, CIFAR10, and GTSRB (for networks and data see Figs. 1, 3d, A3, and A2). **a-c:** Average classification results on MNIST over computation time measured in the number of update steps of networks with skip + recurrent (SR, a), with skip (S, b), and only feed-forward (FF, c) connections. In a), scaling of the abscissa changes at the vertical dashed line for illustration purposes. **d:** The input (top row) is composed of digits (bottom row) and noise (middle row). Note that the input is aligned to the time axis in (a). Red diamonds and blue stars indicate inputs sampled by streaming and sequential rollouts, respectively. **e:** Classification results of the network DSR2 on CIFAR10. **f:** Accuracies at time of first output of sequential rollout (see (V) in e) over networks DSR0 - DSR6 (red and blue curves; left axis). Differences of first response times between streaming and sequential rollouts (see (IV) in e; black dotted curve; right axis). **g:** Average accuracies on GTSRB sequences starting at index 0 of the original sequences. **h:** Final classification accuracies (see (VI) in g) over the start index of the input sequence. Standard errors are shown in all plots except e and f and are too small to be visible in (a-c).

the preceding rollout window (see discussion Sec. 5). Hence, the inference factor of a rollout pattern is used to determine the number of update steps between responses (see $F(R^{\mathrm{str}})$, $F(R^{\mathrm{seq}})$ in Fig. 3a).

**Datasets:** Rollout patterns are evaluated on three datasets: MNIST [50], CIFAR10 [51], and the German traffic sign recognition benchmark (GTSRB) [52]. To highlight the differences between different rollout patterns, we apply noise (different sample for each frame) to the data (see Fig. 3d and Fig. A3b, c). In contrast to data without noise, a single image is now not sufficient for a good classification performance anymore and temporal integration is necessary. In case of GTSRB, this noise can be seen as noise induced by the sensor as predominant under poor lighting conditions. GTSRB contains tracks of 30 frames from which sections are used as input sequences.

**Networks:** We compare the behavior of streaming and sequential rollout patterns on MNIST for three different networks with two hidden layers (FF, S, SR; see Fig. 1 and Fig. A2). For evaluation on CIFAR10, we generate a sequence of 7 incrementally deeper networks (DSR0 - DSR6, see Fig. A3a) by adding layers to the blocks of a recurrent network with skip connections in a dense fashion (details in Fig. A3a). For evaluation on GTSRB, we used DSR4 leaving out the recurrent connection. Details about data, preprocessing, network architectures, and the training process are given in Sec. A2.

**Results:** Rollouts are compared on the basis of their test accuracies over the duration (measured in update steps) needed to achieve these accuracies (Fig. 3a-c, e, and g).

We show behavioral differences between streaming and sequential rollouts for increasingly complex networks on the MNIST dataset. In the case of neither recurrent, nor skip connections (see FF in

Fig. A2), the streaming rollout is mathematically identical to the sequential rollout. Neither rollout can integrate information over time and, hence, both perform classification on single images with the same response time for the first input image and same accuracy (see Fig. 3c). However, due to the pipelined structure of computations in the streaming case, outputs are more frequent.

For networks with skip, but without recurrent connections (see S in Fig. A2), the behavioral difference between streaming and sequential rollouts can be shown best. While the sequential rollout still only performs classification on single images, the streaming rollout can integrate over two input images due to the skip connection that bridges time (see Fig. 3b).

In the streaming case, skip connections cause shallow shortcuts in time that can result in earlier (see (I) in Fig. 3a), but initially worse performance than for deep sequential rollouts. The streaming rollout responds 1 update step earlier than the sequential rollout since its shortest path is shorter by 1 (see Fig. 2). These early first estimations are later refined when longer paths and finally the longest path from input to output contribute to classification. For example, after 3 time steps in Fig. 3a, the streaming rollout uses the full network. This also applies to the sequential rollout, but instead of integrating over two images (frames 0 and 1), only the image of a single frame (frame 1) is used (cf. blue to red arrows connecting Fig. 3d and a).

Due to parallel computation of the entire frame in the streaming case, the sampling frequency of input images (every time step; see red diamonds in Fig. 3d) is maximal ($F(R^{\text{str}}) = 1$ in Fig. 3a; see d in Theorem 1 in Sec. 3). In contrast, the sampling frequency of the sequential rollout decreases linearly with the length of the longest path ($F(R^{\text{seq}}) = 3$ in Fig. 3a; blue stars in Fig. 3d).

High sampling frequencies and shallow shortcuts via skip connections establish a high degree of temporal integration early on and result in better early performance (see (II) in Fig. 3a). In the long run, however, classification performances are comparable between streaming and sequential rollouts and the same number of input images is integrated over (see (III) in Fig. 3a).

We repeat similar experiments for the CIFAR10 dataset to demonstrate the increasing advantages of the streaming over sequential rollouts for deeper and more complex networks. For the network DSR2 with the shortest path of length 4 and longest path of length 6, the first response of the streaming rollout is 2 update steps earlier than for the sequential rollout (see (IV) in Fig. 3e) and shows better early performance (see (V) in Fig. 3e). With increasing depth (length of the longest path) over the sequence of networks DSR0 - DSR6 (see Fig. A3a), the time to first response stays constant for streaming, but linearly grows with the depth for sequential rollouts (see Fig. 3f black curve). The difference of early performance (see (V) in Fig. 3e) widens with deeper networks (Fig. 3f).

For evaluation of rollouts on GTSRB, we considere the DSR4 network. Self-recurrence is omitted since the required short response times of this task cannot be achieved with sequential rollouts due to the very small sampling frequencies. Consequently, for fair comparison, we calculate the classifications of the first 8 images in parallel for the sequential case. In this case, where both rollouts use the same amount of computations, performance for the sequential rollout increases over time due to less blurry input images, while the streaming rollout in addition performs temporal integration using skip connections and yields better performance (see (VI) in Fig. 3g). This results in better performance of streaming compared to sequential rollouts for more distant objects (Fig. 3h).

## 5   Discussion and Conclusion

The presented theory for network rollouts is generically applicable to a vast variety of deep neural networks (see Sec. A1.2) and is not constrained to recurrent networks but could be used on forward (e.g., VGG [53], AlexNet [54]) or skipping networks (e.g., ResNet [1], DenseNet [55]). We restricted rollout patterns to have values $R(e) \in \{0, 1\}$ and did neither allow edges to bridge more than 1 frame $R(e) > 1$ nor pointing backwards in time $R(e) < 0$. The first case is subsumed under the presented theory using copy-nodes for longer forward connections, and for $R(e) < 0$ rollouts with backward connections loose the real-time capability, because information from future frames would be used.

In this work, we primarily investigated differences between rollout patterns in terms of the level of parallelization they induce in their rollout windows. But using different rollout patterns is not a mere implementation issue. For some networks, all rollout patterns yield the same mathematical behavior (e.g., mere feed-forward networks without any skip or recurrent connections, cf. Fig. 3c). For other networks, different rollout patterns (see Sec. 3) may lead to differences in the behavior of their rollout

windows (e.g., Fig. 3b). Hence, parameters between different rollout patterns might be incompatible. The theoretical analysis of *behavioral* equivalency of rollout patterns is a topic for future work.

One disadvantage of the streaming rollout pattern seems to be that deeper networks also require deeper rollout windows. Rollout windows should be at least as long as the longest minimal path connecting input to output, i.e., all paths have appeared at least once in the rollout window. For sequential rollout patterns this is not the case, since, e.g., for a feed-forward network the longest minimal path is already contained in the first frame. However, for inference with streaming rollouts instead of using deep rollouts we propose to use shallow rollouts (e.g., $W = 1$) and to initialize the zero-th frame of the next rollout window with the last (i.e., first) frame of the preceding rollout window. This enables a potentially infinite memory for recurrent networks and minimizes the memory footprint of the rollout window during inference.

Throughout the experimental section, we measured runtime by the number of necessary update steps assuming equal update time for every single node update. Without this assumption and given fully parallel hardware, streaming rollouts still manifest the best case scenario in terms of maximal parallelization and the inference of a single frame would take the runtime of the computationally most expensive node update. However, sequential rollouts would not benefit from the assumed parallelism of such hardware and inference of a single frame takes the summed up runtime of all necessary node updates. The streaming rollout favors network architectures with many nodes of approximately equal update times. In this case, the above assumtion approximately holds.

The difference in runtime between rollout patterns depends on the hardware used for execution. Although commonly used GPUs provide sufficient parallelism to speed up calculations of activations within a layer, they are often not parallel enough to enable the parallel computation of multiple layers. Novel massively parallel hardware architectures such as the TrueNorth chip [56, 57] allow to store and run the full network rollouts on-chip reducing runtime of rollouts drastically and therefore making streaming rollouts highly attractive. The limited access to massively parallel hardware may be one reason, why streaming rollouts have not been thoroughly discussed, yet.

Furthermore, not only the hardware, but also the software frameworks must support the parallelization of independent nodes in their computation graph to exploit the advantages of streaming rollouts. This is usually not the case and by default sequential rollouts are used. For the experiments presented here, we use the Keras toolbox to compare different rollout patterns. To realize arbitrary rollout patterns in Keras, instead of using Keras' build-in RNN functionalities, we created a dedicated model builder which explicitly generates the rollout windows. Additionally, we implemented an experimental toolbox (Tensorflow and Theano backends) to study (define, train, evaluate, and visualize) networks using the streaming rollout pattern (see Sec. A3). Both are available as open-source code[3].

Similar to biological brains, synchronization of layers (nodes) plays an important role for the streaming rollout pattern. At a particular time (frame), different nodes may carry differently delayed information with respect to the input. In this work, we evaluate network accuracy dependent on the delayed response. An interesting area for future research is the exploration of mechanisms to guide and control information flow in the context of the streaming rollout patterns, e.g., through gated skips (bottom-up) and recurrent (top-down) connections. New sensory information should be distributed quickly into deeper layers. and high-level representations and knowledge of the network about its current task could stabilize, predict, and constrain lower-level representations.

A related concept to layer synchronization is that of *clocks*, where different layers, or more generally different parts of a network, are updated with different frequencies. In this work, all layers are updated equally often. In general, it is an open research question to which extend clocking and more generally synchronization mechanisms should be implicit parts of the network and hence learnable or formulated as explicit a-priory constraints.

**Conclusion:** We presented a theoretical framework for network rollouts and investigated differences in behavior and model-parallelism between different rollouts. We especially analysed the streaming rollout, which fully disentangles computational dependencies between nodes and hence enables full model-parallel inference. We empirically demonstrated the superiority of the streaming over non-streaming rollouts for different image datasets due to faster first responses to and higher sampling of inputs. We hope our work will encourage the scientific community to further study the advantages and behavioral differences of streaming rollouts in preparation to future massively parallel hardware.

**Acknowledgments**

The authors would like to thank Bastian Bischoff, Dan Zhang, Jan-Hendrik Metzen, and Jörg Wagner for their valuable remarks and discussions.

## Footnotes

[1] given the state of all input nodes at all frames and initial states for all nodes at the zero-th frame

[2]Here, the most sequential rollout is unique since the used networks do not contain cycles of length greater 1. For sequential rollouts that are ambiguous see bottom row of Fig. 2.

[3]`https://github.com/boschresearch/statestream`

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
