[Supplementary Material]

**The streaming rollout of deep networks - towards fully model-parallel execution**
**Supplementary material**

## A1    Proofs and notes for theory chapter

To improve readability, we will restate certain parts of the theory chapter from the main text.

### A1.1    Examples for networks following our definition in Sec. 3

Figure A1: Examples of networks covered by the presented theory in Sec. 3. The crossed-out network has no input and is consequently not a network by our definition.

### A1.2    Note: Connecting the theory of networks with deep neural networks

For deep networks, nodes $v$ correspond to layers and edges $e$ to transformations between layers, such as convolutions (e.g., see networks in Fig. A1 and Fig. A2). To a node $v$ a state $x_v \in \mathbb{R}^{D_v}$ is assigned, e.g. an image with $D_v = 32 \times 32 \times 3$. Let denote $y_e$ the result of the transformation $f_e$ which is specified by an edge $e = (u, v)$:

$$y_e = f_e(\theta_e, x_u),$$

where $\theta_e$ are parameters of the edge $e$, e.g. a weight kernel.

For the node $v$, let $\text{SRC}_v$ denote the set of all edges targeting $v$. Ignoring the temporal dimension, a node's state is then computed as:

$$x_v = f_v(\vartheta_v, y_{e_v^1}, \ldots, y_{e_v^{|\text{SRC}_v|}}),$$

where $\vartheta_v$ are parameters of the vertex $v$ (e.g., biases), and the mapping $f_v$ specifying how the sources are combined (e.g., addition and/or multiplication). Most architectures and network designs can be subsumed under this definition of a network, because we do not impose any constraints on the node and edge mappings $f_v, f_e$.

For the experiments in this work, for every node $v$ all results of incoming transformations were summed up:

$$x_v = f_v(b, y_{e_v^1}, \ldots, y_{e_v^{|\text{SRC}_v|}}) = \sigma \left( b + \sum_{e \in \text{SRC}_v} y_e \right),$$

where $\sigma$ is some activation function, $b$ is a channel-wise bias and the edge transformations $f_e$ were convolutions with suitable stride to provide compatible dimensions for summation.

### A1.3    Proofs for definition (rollout) in Sec. 3

Let $N = (V, E)$ be a network. We call a mapping $R : E \to \{0, 1\}$ a **rollout pattern** of $N$. For a rollout pattern $R$, the **rollout window** of size $W \in \mathbb{N}$ is the directed graph $R_W = (V_W, E_W)$ with:

$$
\begin{aligned}
V_W &:= \{0, \ldots, W\} \times V, \quad \overline{v} = (i, v) \in V_W \\
E_W &:= \{((i, u), (j, v)) \in V_W \times V_W \mid (u, v) \in E \ \wedge \ j = i + R((u, v))\}.
\end{aligned}
\tag{8}
$$

We dropped the dependency of especially $E_W$ on the rollout pattern $R$ in the notation. A rollout pattern and its rollout windows are called **valid** iff $R_W$ is acyclic for one and hence for all $W \in \mathbb{N}$. We denote the set of all valid rollout patterns as $\mathcal{R}_N$, and the rollout pattern for which $R \equiv 1$ the **streaming rollout** $R^{\text{stream}} \in \mathcal{R}_N$. We say two rollout patterns $R$ and $R'$ are **equally model-parallel** iff for all edges $e = (u, v) \in E$ not originating in the network's input $u \notin I_N$ are equal $R(e) = R'(e)$. For $i \in \{0, \dots, W\}$, the subset $\{i\} \times V \subset V_W$ is called the $i$-th **frame**.

**Note (interpretation):** Here, we show how this definition reflects the intuition, that a rollout should be consistent with the network in the sense that it should *contain* all edges / nodes of the network and should not add new edges / nodes, which are not present in the network. Further, we show that this definition yields rollout windows which are temporally consistent and that rollout windows are consistent with regards to each other:

- **Rollout windows cannot add *new* edges / nodes:** By this, we mean, that a rollout window only contains derived nodes and edges from the original network and for example cannot introduce edges between nodes in the rollout window, which were not already present in the network. This follows directly from the definition of $E_W$.

- **Edges / nodes of the network are contained in a rollout window:** For vertices this is trivial and for edges $e = (u, v) \in E$ always $((0, u), (R(e), v)) \in E_W$.

- **Rollout windows contain no temporal backward edges:** A backward edge is an edge $((i, u), (j, v)) \in E_W$ with $j < i$. But we know for all edges that $j = i + R((u, v))$.

- **Temporal consistency:** Temporal consistency means that for an edge $((i, u), (j, v)) \in E_W$ and a second edge between the *same* nodes $((i_\star, u), (j_\star, v)) \in E_W$ the temporal gap is the same $j - i = j_\star - i_\star$. By definition, both are equal to $R((u, v))$.

- **Rollout windows are compatible with each other:** We show that $R_W$ is a sub-graph of $R_{W+1}$, in the sense that $V_W \subset V_{W+1}$ and $E_W \subset E_{W+1}$: From the definition, this is obvious for the set of vertices and edges, but nevertheless we will state it for edges anyway: Let $\bar{e} \in E_W$ with $\bar{e} = ((i, u), (j, v))$. Then by definition $(u, v) \in E$ and $j = i + R((u, v))$. Hence, $\bar{e} \in R_{W+1}$.

**Proof (definition of valid rollout pattern is well-defined):** For a rollout pattern, we prove that if the rollout window of a certain size $W$ is valid, then the rollout window for any size is valid: Let $R$ be a rollout pattern for a network $N$ and $R_W$ be a valid rollout window. Because $R_W$ hence contains no cycles, also $R_{W'}$ for $W' < W$ contains no cycles (see statement about rollout window compatibility from above). Using induction, it is sufficient to show that $R_{W+1}$ is valid. Assuming it is not, let $p$ be a cycle in $R_{W+1}$. Because there are no temporal backward edges (see above) $p$ has to be contained in the last, the $(W + 1)$-th frame. Because of the temporal consistency of rollout windows (see above), there are now cycles in all previous frames which contradicts the validity of $R_W$.

**Proof (streaming rollout exists and is valid):** The streaming rollout pattern $R^{\text{stream}} \equiv 1$ always exists, because according to our network definition, $E$ is not empty. Further, the streaming rollout pattern is always valid: Assuming that this is not the case, let $R_W^{\text{stream}}$ be a rollout window of size $W$ which is not acyclic and let $p$ be a cycle in $R_W^{\text{stream}}$. Because there are no backward edges $\bar{e} = ((i, u), (j, v)) \in E_W$ with $j < i$, all edges of the cycle must be inside a single frame, which is in contradiction to $R^{\text{stream}} \equiv 1$.

**Note (streaming rollout is un-ambiguous):** Considering the sets of all *most streaming* and *most non-streaming* rollout patterns

$$R_{\text{streaming}} = \left\{ R \in \mathcal{R}_N \ \middle| \ |R^{-1}(1)| = \max_{R_\star \in \mathcal{R}_N} |R_\star^{-1}(1)| \right\}$$

$$R_{\text{non-streaming}} = \left\{ R \in \mathcal{R}_N \ \middle| \ |R^{-1}(0)| = \max_{R_\star \in \mathcal{R}_N} |R_\star^{-1}(0)| \right\}$$

we have shown above that $|R_{\text{streaming}}| = 1$ and this is exactly the streaming rollout. In contrast, $|R_{\text{non-streaming}}| \geq 1$ especially for networks containing cycles with length greater 1. In this sense, the streaming rollout is un-ambiguous because it always uniquely exists while *the* most-sequential rollout is ambiguous.

## A1.4 Proof for Lemma 1 in Sec. 3

**Lemma 1:** Let $N = (V, E)$ be a network. The number of valid rollout patterns $|\mathcal{R}_N|$ is bounded by:

$$1 \leq n \leq |\mathcal{R}_N| \leq 2^{|E|-|E_{\text{rec}}|}, \tag{9}$$

where $E_{\text{rec}}$ is the set of all self-connecting edges $E_{\text{rec}} := \{(u, v) \in E \mid u = v\}$, and $n$ either:

- $n = 2^{|E_{\text{forward}}|}$, with $E_{\text{forward}}$ being the set of edges not contained in any cycle of $N$, or
- $n = \prod_{p \in C} (2^{|p|} - 1)$, $C \subset C_N$ being any set of minimal and pair-wise non-overlapping cycles.

**Proof** $|\mathcal{R}_N| \leq 2^{|E|-|E_{\text{rec}}|}$**:**   The number of all (valid and invalid) rollout patterns is $2^{|E|}$, because the pattern can assign 0 or 1 to every edge. In order to be valid (acyclic rollout windows), the pattern has to assign 1 at least to every self-connecting edge.

**Proof** $1 \leq n$**:**   Concerning the forward case: According to the definition of a network, $I_N$ is not empty and hence there always exists at least one forward edge $|E_{\text{forward}}| > 0$. Concerning the recurrent case: It is easy to see that $n$ is greater than 0, increases with $|C|$ and that $C$ has to be at least the empty set.

**Proof** $n \leq |\mathcal{R}_N|$ **forward case:**   Considering the streaming rollout pattern $R^{\text{stream}} \equiv 1$ which always exists and is always valid (see above), we combinatorially can construct $2^{|E_{\text{forward}}|}$ different valid rollout patterns on the basis of the streaming rollout pattern by combinatorially changing $R(e)$ for all forward edges $e \in E_{\text{forward}}$.

**Proof** $n \leq |\mathcal{R}_N|$ **recurrent case:**   W.l.o.g. in case $C_N = \emptyset$ we set $n = 1$. Otherwise let $C \subset C_N$ be any set of minimal and pair-wise non-overlapping cycles. Based on the streaming rollout pattern we will again construct the specified number of rollout patterns. The idea is that every cycle $p \in C$ gives rise to $2^{|p|} - 1$ different rollout patterns by varying the streaming rollout $R^{\text{stream}}(E) \equiv 1$ on all edges in $p$ and we have to subtract the one rollout for which $R(p) \equiv 0$, because for this specific rollout pattern, the cycle $p$ does not get unrolled. Because the cycle is minimal, those $2^{|p|} - 1$ patterns are different from one another. Because all cycles in $C$ are disjunct we can combinatorially use this construction across all cycles of $C$ and constructed $\prod_{p \in C} (2^{|p|} - 1)$ valid rollouts.

## A1.5 Proof update steps convergence to full state in Sec. 3

Let $R_W$ be a rollout window for a valid rollout pattern $R$ of the network $N = (V, E)$. Then, starting from the initial state $S_{\text{init}}$ and successively applying update steps $U$, converges always to the full state $S_{\text{full}}$:

$$\exists n \in \mathbb{N} : U^n(S_{\text{init}}) = S_{\text{full}}$$

**Proof:**   Using induction, we show this without loss of generality for $R_1$. Assuming that this is not the case, then there exists a state $S \in \Sigma_1$, such that

$$\forall n \in \mathbb{N} : U^n(S) = S, \text{ and } \exists \overline{v} = (1, v) \in V_1 : S(\overline{v}) = 0$$

But being unable to update $\overline{v}$ means, that there is another node that is input to $\overline{v}$ which is also not updated yet $(1, v_1) \in V_1$ and $S((1, v_1)) = 0$. Because there are no loops in $R_1$ these nodes are not the same $v \neq v_1$. This line of argument can now also be applied to $v_1$ leading to a third node $(1, v_2)$ with $S((1, v_2)) = 0$ and $v \neq v_1 \neq v_2$ and so on. Because we only consider networks with $|V| < \infty$ this leads to a contradiction.

## A1.6 Proof Definition of inference tableau in Sec. 3

For a valid rollout pattern $R$ and a rollout window $R_W$, we defined the inference tableau as the mapping $T : V_W \to \mathbb{N}$ with:

$$T(\overline{v}) := \max_{p \in P_{\overline{v}}} |p| = \operatorname*{argmin}_{n \in \mathbb{N}} \{U^n(S_{\text{init}})(\overline{v}) = 1\}$$

For this, we have to show, that the equation holds.

**Proof:** We denote:

$$T^{\max}(\overline{v}) := \max_{p \in P_{\overline{v}}} |p|$$

$$T^{\min}(\overline{v}) := \operatorname*{argmin}_{n \in \mathbb{N}} \{ U^n(S_{\mathrm{init}})(\overline{v}) = 1 \}$$

and have to show $T^{\min} \equiv T^{\max}$. The proof is divided into two parts, first showing that the number of necessary update steps to update a certain node $\overline{v}$ is higher or equal the length of any path $p \in P_{\overline{v}}$ and hence $T^{\min} \geq T^{\max}$. In the second part of the proof, we show that maximal paths $p \in P_{\overline{v}}$ get successively updated at every update step.

In the first part, we will prove the following statement: For every $\overline{v} \in V_W$ and $p \in P_{\overline{v}}$:

$$T^{\min}(\overline{v}) \geq T^{\min}(p(1)_{\mathrm{src}}) + |p|. \tag{10}$$

Here, we denoted again the edges of the path as $p(i) = (p(i)_{\mathrm{src}}, p(i)_{\mathrm{tgt}}) \in E_W$. In words this means, that for every path in a valid rollout window, the tableau values of the paths first $p(1)_{\mathrm{src}}$ and last $\overline{v} = p(|p|)_{\mathrm{tgt}}$ node differ at least about the length of the path. This is clear for paths of length one $|p| = 1$, because $p(1)_{\mathrm{tgt}}$ can neither be updated before nor at the same update step as $p(1)_{\mathrm{src}}$, because $p(1)_{\mathrm{src}}$ is an input of $p(1)_{\mathrm{tgt}}$. Using induction and the same argument for paths of greater lengths $|p| = n$ proves (10) and therefore also $T^{\min} \geq T^{\max}$.

In the second part of the proof, we will show that for all $\overline{v} \in V_W$ all paths $p \in P_{\overline{v}}$ of maximal length get updated node by node in each update step:

$$U^{i-1}(S_{\mathrm{init}})(p(i)_{\mathrm{tgt}}) = 0$$

$$U^{i}(S_{\mathrm{init}})(p(i)_{\mathrm{tgt}}) = 1$$

for $i \in \{1, \ldots, |p|\}$.

We will prove this via induction over maximal path lengths. For $\overline{v} \in V_W$ for which the maximum length of a path $p \in P_{\overline{v}}$ is zero $|p| = 0$ and hence $P_{\overline{v}} = \emptyset$ we know by definition of $P_{\overline{v}}$ and because the rollout window is connected to the initial state (see Sec. A1.5) that $U^0(S_{\mathrm{init}})(\overline{v}) = S_{\mathrm{init}}(\overline{v}) = 1$. This proves the second part for $\overline{v}$ with maximum path length zero. Now we consider $\overline{v} \in V_W$ for which the maximum length of a path $p \in P_{\overline{v}}$ is one $|p| = 1$. Because $p$ is maximal, its first node is in the initial state $S_{\mathrm{init}}(p(1)_{\mathrm{src}}) = 1$ and due to the definition of $P_{\overline{v}}$ it is $S_{\mathrm{init}}(p(1)_{\mathrm{tgt}}) = 0$. Further, because $p$ is maximal and of length 1, the initial state of all inputs to $p(1)_{\mathrm{tgt}}$ is 1 and hence $p(1)_{\mathrm{tgt}}$ can be updated in the first update step $U(S_{\mathrm{init}})(p(1)_{\mathrm{tgt}}) = 1$. This proves the second part for $\overline{v}$ with maximum path length one.

Let now be $n \geq 2$, and we assume that the statement is true for nodes $\overline{v}$ for which maximal paths $p \in P_{\overline{v}}$ have length $n$. Be $\overline{v}$ now a node in $V_W$ for which the maximal length of a path $p \in P_{\overline{v}}$ is $n+1$. If the end node of a maximal path $p \in P_{\overline{v}}$ cannot be updated $U^{n+1}(S_{\mathrm{init}})(p(n+1)_{\mathrm{tgt}}) = 0$, then one of this end node's inputs $\overline{v}_{\mathrm{input}} \in V_W$ was not yet updated $U^n(S_{\mathrm{init}})(\overline{v}_{\mathrm{input}}) = 0$. But because $p$ is maximal and of length $n+1$, and $\overline{v}_{\mathrm{input}}$ is input to $\overline{v}$, the maximum length of paths in $P_{\overline{v}_{\mathrm{input}}}$ is $n$. Hence $U^n(S_{\mathrm{init}})(\overline{v}_{\mathrm{input}}) = 1$ contradicting that $\overline{v}_{\mathrm{input}}$ was not yet updated and therefore proving the second part of the proof. This proves $T^{\min} \equiv T^{\max}$ and hence both can be used to define the inference tableau.

### A1.7 Proof for Theorem 1 in Sec. 3

**Theorem 1:** Let $R$ be a valid rollout pattern for the network $N = (V, E)$ then the following statements are equivalent:

a) $R$ and the streaming rollout pattern $R^{\mathrm{stream}}$ are equally model-parallel.

b) The first frame is updated entirely after the first update step: $F(R) = 1$.

c) For $W \in \mathbb{N}$, the $i$-th frame of $R_W$ is updated at the $i$-th update step:

$$\forall (i, v) \in V_W \ : \ T((i, v)) \leq i.$$

d) For $W \in \mathbb{N}$, the inference tableau of $R_W$ is minimal everywhere and over all rollout patterns (most frequent responses & earliest response):

$$\forall \overline{v} \in V_W \ : \ T_{R_W}(\overline{v}) = \min_{R' \in \mathcal{R}_N} T_{R'_W}(\overline{v}).$$

**Proof:** Equivalency of statements a) - d) will be shown via a series of implications connecting all statements:

**a) $\implies$ b):** Assuming there is a $\overline{v} = (1, v)$ which cannot be updated with the first update step, then there has to be an input $(1, v_{\text{input}})$ of $\overline{v}$ for which $S_{\text{init}}((1, v_{\text{input}})) = 0$ which contradicts that $R$ is equally model-parallel to the streaming rollout.

**b) $\implies$ a):** Assuming $R(e) = 0$ for an edge $e = (u, v) \in E$ with $u \notin I_N$, would yield a dependency of $(1, v)$ on $(1, u)$. Because $u \notin I_N$, $(1, u)$ is not updated at the beginning $S_{\text{init}}((1, u)) = 0$ and therefore $U^1(S_{\text{init}})((1, v)) = 0$ and hence $T((1, v)) \geq 2$ which contradicts b).

**c) $\implies$ b):** Trivial.

**a) $\implies$ c):** Let $\overline{v} = (i, v) \in V_W$. First we note, that every maximal path $p \in P_{\overline{v}}$ has to start in the initial state $S_{\text{init}}(p(1)_{\text{src}}) = 1$, otherwise we can extend $p$ to a longer path. We will use the definition of $T$ over maximum path lengths to prove c). Let $R$ be equally model-parallel to the streaming rollout and $p \in P_{\overline{v}}$ a path of maximal length. We know $S_{\text{init}}(p(1)_{\text{src}}) = 1$ and hence either $p(1)_{\text{src}} \in \{0\} \times V$ or $p(1)_{\text{src}} \in \{0, \dots, W\} \times I_N$. For the first case, it is easy to see that $|p| = i$, because $R$ is equally model-parallel to the streaming rollout and hence one frame is bridged $R(e) = 1$ for every edge $e$ in $p$. For the second case $p(1)_{\text{src}} \in \{0, \dots, W\} \times I_N$, it follows from the same argument as before that $|p| = i - i_{\text{src}}$ with $p(1)_{\text{src}} = (i_{\text{src}}, v_{\text{src}})$ which proves c).

**a) $\implies$ d):** For this proof we introduce **induced paths**: Let $R$ be a valid rollout pattern, $\overline{v} = (i, v) \in R_W$ and $p_R \in P_{\overline{v}}^{R_W}$ (same as $P_{\overline{v}}$ from rollout definition but now expressing the dependency on the rollout window $R_W$):

$$\begin{aligned} p_R(k) &= \overline{e}^k \\ &= ((j_{\text{src}}^k, e_{\text{src}}^k), (j_{\text{tgt}}^k, e_{\text{tgt}}^k)) \\ &= ((j_{\text{src}}^k, e_{\text{src}}^k), (j_{\text{src}}^k + R(e^k), e_{\text{tgt}}^k)), \end{aligned}$$

for $k \in \{1, \dots, |p_R|\}$ and $e^k = (e_{\text{src}}^k, e_{\text{tgt}}^k) \in E$. Let $R'$ be a second valid rollout pattern and let denote $n = |p_R|$. Notice that $(j_{\text{tgt}}^n, e_{\text{tgt}}^n) = (i, v)$. We want to define the induced path $p_{R'} \in P_{\overline{v}}^{R'_W}$ as the path also ending at $\overline{v} \in R'_W$, backwards using the *same* edges as $p_R$ and respecting the rollout pattern $R'$. We define this induced path $p_{R'} \in P_{\overline{v}}^{R'_W}$ of $p_R$ recursively, beginning with the last edge of $p_R$, as the end of the following sequence of paths, starting with the path:

$$\begin{aligned} p_{R',1} &: \{1\} \to E_{R'_W} \\ p_{R',1}(1) &= ((i - R'(e^n), e_{\text{src}}^n), (i, e_{\text{tgt}}^n)) \end{aligned}$$

Recursively we define:

$$\begin{aligned} p_{R',m} &: \{1, \dots, m\} \to E_{R'_W} \\ p_{R',m}(k) &= p_{R',m-1}(k-1), \ k \in \{2, \dots, m\} \\ p_{R',m}(1) &= ((i - s_{R',p_R}(m), v_{\text{src}}^{n-m+1}), (i - s_{R',p_R}(m-1), v_{\text{tgt}}^{n-m+1})) \end{aligned}$$

with $s_{R',p_R}(m) = \sum_{k=1}^{m} R'(e^{n-k+1})$. In words, $s_{R',p_R}(m)$ is the *frame length* of the last $m$ edges of the path $p_R$ under the rollout pattern $R'$. The sequence stops at a certain $m$, either if no edges are left in $p_R$: $m = n$ or at the first time the source of the path's first edge reaches the 0-th frame: $i - s_{R',p_R}(m) = 0$. With this definition we can proceed in the prove of a) $\implies$ d):

Let $R$ be equally model-parallel to the streaming rollout pattern, $W \in \mathbb{N}$, and $\overline{v} \in V_W$. Let further be $p_R \in P_{\overline{v}}^{R_W}$ a path of maximal length, $R'$ be any valid rollout pattern, and $p_{R'}$ be the induced path of $p_R$. We want to show that $|p_R| = |p_{R'}|$.

If both rollouts are equally model-parallel on the edges of the path $\{e^1, \dots, e^{|p_R|}\}$ (this means $R(e^k) = R'(e^k)$ for $k \in \{1, \dots, |p_R|\}$ if $e^1$ does not originate in the input $e_{\text{src}}^1 \notin I_N$, and for

$k \in \{2, \ldots, |p_R|\}$ if $e^1$ does originate in the input), the path $p_R$ and its induced path $p_{R'}$ are the same up to their first edge which might or might not bridge a frame, but in both cases $|p_R| = |p_{R'}|$.

If the rollouts are not model-parallel on the edges of the path and hence differ on at least one edge $e^k$ which does not originate in the input, and because $R$ is equally model-parallel to the streaming rollout, it is:

$$s_{R,p_R}(|p_R|) > s_{R',p_R}(|p_{R'}|). \tag{11}$$

Because the induced path using the same rollout cannot loose length, we also know:

$$i - s_{R,p_R}(|p_R|) \geq 0. \tag{12}$$

Greater than zero would be the case for $p_R$ originating in the input $p_R(1)_{\text{src}} \in \{1, \ldots, W\} \times I_N$. Combining (11) and (12) yields:

$$i - s_{R',p_R}(|p_{R'}|) > 0.$$

Considering the two stopping criteria from the sequence of paths used to define the induced path from above, this proves $|p_R| = |p_{R'}|$.

We now have proven that the induced path $p_{R'}$ from a maximal path $p_R$ in a rollout window from a rollout pattern $R$ which is equally model-parallel to the streaming rollout is never shorter than $p_R$ (especially for highly sequential $R'$, most $p_{R'}$ are not of maximal length). This means, that the maximal length of paths in $P_{\overline{v}}^{R'_W}$ is at least as large as the maximal length of paths in $P_{\overline{v}}^{R_W}$ which by definition of the inference tableau proves a) $\implies$ d).

**d) $\implies$ b):**   Trivial.

## A2   Details about networks, data, and training

In the depiction of network architectures (Fig. 1, Fig. A3, and Fig. A2), connections between nodes are always realized as convolutional or fully connected layers. In case a node (layer) is the target of several connections, its activation is always computed as the sum over outputs of these connections. This is mathematically equivalent to concatenating all inputs of the layer and applying a single convolution on the concatenation.

Figure A2: Neural networks (gray boxes) used for MNIST (Fig. 3a-c) with different rollouts. Schematics of a feed-forward network (FF, **a**, green) with its corresponding sequential (**b**, blue) and streaming (**c**, red) rollouts. Nodes represent layers, edges represent transformations, e.g. convolutions. Only one rollout step is shown and each column in (b) and (c) is one frame within the rollout. Rollouts are also shown for networks with an additional skip connection (S, **d-f**). Node labels on the left are referred to in Sec. A2.

**MNIST**   The network designs are shown in Fig. 1 and Fig. A2. The size of the layers (pixels, pixels, features) are: input image I with (28, 28, 1), hidden layer H1 with (7, 7, 16), hidden layer H2 with (1, 1, 128) and output layer O with (1, 1, 10).

The following network design specifications were applied with A-B meaning the edge between layer A and layer B. Some of these edges only exist in the networks with skip connection (S) or with skip and self-recurrent connections (SR). For node labels see Fig. A2:

- I-H1: a convolution with receptive field 7 and stride 4

- H1-H2 and H2-O: fully connected layers
- H1-O: a fully connected layer
- H1-H1-recurrence: a convolution with receptive field 3 and stride 1

Figure A3: **a:** A selection of the sequence of networks evaluated on CIFAR10 (for details see Sec. A2). For evaluating the GTSRB dataset the network DSR4 is used, but without the self-connection of node H1. The input of the networks are images with added Gaussian noise as shown in **b:** for CIFAR10 and **c:** for GTSRB (for details see Table A1).

**CIFAR10**   The network design is shown in Fig. A3a. We used a sequence of 7 increasingly deep network architectures with the first network DSR0 being a simple (3 hidden layers) forward design and the first hidden layer having a self-recurrent connection. We added additional hidden layers to generate the next networks in the following way: H11 to DSR0, H21 to DSR2, H12 to DSR3, ..., H23 to DSR6. Note that every network is a sub-network of its successor. Hence, the length of the shortest path is always 4, while the length of the longest path increases from 4 to 11 by 1 for every consecutive network.

The size of the layers (pixels, pixels, features) are: input image I with $(32, 32, 3)$ and hidden layers H1, H11, H12, H13 with $(32, 32, 32)$ and H2, H21, H22, H23 with $(16, 16, 64)$, fully connected layer HD with $(4, 4, 512)$ and output layer O with $(1, 1, 10)$.

The following network design specifications were applied:

- I-H1: a convolution with receptive field 5 and stride 1
- H1-H11, H11-H12, H12-H13: a convolution with receptive field 3 and stride 1
- H2-H21, H21-H22, H22-H23: a convolution with receptive field 3 and stride 1
- H13-H2: convolutions with receptive field 3 and stride 2
- H23-HD: convolutions with receptive field 3 and stride 4
- H1-H1-recurrence: a convolution with receptive field 3 and stride 1
- skip connections H1-H12, H1-H13, H1-H2, H11-H13, H11-H2, H12-H2 and H2-H22, H2-H23, H2-HD, H21-H23, H21-HD, H22-HD: convolution with receptive field 3 and stride $\frac{\text{input size}}{\text{output size}}$

**GTSRB**   For the experiments, the network DSR4 shown in Fig. A3a was used without the self-recurrence H1-H1 connection. Design specifications are adapted from the CIFAR10 networks with input image I with $(32, 32, 3)$ and output layer O with $(1, 1, 43)$. For each repetition, 80% of the data was randomly taken for training, 10% for validation, and 10% for testing.

**Training details** To train networks, we used RMSprop ([58]) with an initial learning rate of $10^{-4}$ and an exponential decay of $10^{-6}$. All networks were trained for 100 epochs. A dropout rate of $0.25$ was used for all but the last hidden layer, for which a rate of $0.5$ was used. The loss for the rolled-out networks is always the mean over the single-frame prediction losses, for which we used cross-entropy. At the zero-th frame, states of all but the input layers were initialized with zero.

Details about experimental setups and data processing are given in Table A1.

| Data | value range | perturbation | augmentation | training / val. / test size | batch size | reps |
|---|---|---|---|---|---|---|
| Noisy MNIST | [0,1] | 1. $\mathcal{N}(\sigma = 2.0)$; 2. clipped to [0,1] | None | 50k / 10k / 10k | 128 | 6 |
| CIFAR10 | [0,1] | 1. $\mathcal{N}(\sigma = 1.0)$; 2. clipped to [0,1] 3. mean subtracted | horizontal flipping | 40k / 10k / 10k | 64 | 1 |
| GTRSB | [0,1] | 1. $\mathcal{N}(\sigma = 0.5)$; 2. clipped to [0,1] 3. resized to $32 \times 32$ pixels | None | 80% / 10% / 10% of 1305 tracks (30 frames each) | 16 | 12 |

Table A1: Experimental setups for the data sets: Image pixels were scaled (*value range*); then each frame was perturbed adding Gaussian noise with a standard deviation of $\sigma$, clipped back into the value range and for CIFAR10 the channel-wise mean over all training images was subtracted. For GTRSB images of different size were resized. Data *augmentation* was conducted for training and the number of images for training, validation, and testing (*training / val. / test size*) and the *batch sizes* are listed. Experiments were repeated (*reps*) times.

## A3  Toolbox for streaming rollouts

One of the contributions of this work is to provide an open-source toolbox (`https://github.com/boschresearch/statestream`) to design, train, evaluate, and interact with the streaming rollout of deep networks. An example screenshot of the provided user interface is shown in Fig. A4.

Networks are specified in a text file, and a core process distributes the network elements onto separate processes on CPUs and/or GPUs. Network elements are executed with alternating read and write phases, synchronized via a core process, and operate on a shared representation of the network. The toolbox is written in Python and uses the Theano [59] or TensorFlow [60] backend. The shared representation enables parallelization of operations across multiple processes and GPUs on a single machine and enables online interaction.

Figure A4: Visualization example of a simple classification network using the provided toolbox (best viewed in color). The network is shown as graph together with information about the network.