[Reviews · NeurIPS 2018]

Reviewer 1



About my confidence: I'm barely familiar with graph theory and unfamiliar graphical propagation using tableaus etc. I found your arguments to be quite clear and well stated. I'm particularly interested in the concept of response time. (PS: There is a typo on 288-289.) =========================== edit after rebuttal: I agree with Reviewer 3 that simple examples to illustrate each concept would take this paper to the next level. I want to see this work as a longer paper so that the authors do not have to hold back with examples and explanation, analogies etc. In particular it would be useful to link this to some popular rollout papers, like LISTA (this is what I view as the original rollout, which turns iterative algorithm ISTA into an RNN) (Learning Fast Approximationg of Sparse Coding - Yann LeCun et al). However, most rollout papers do not even touch the world outside of numerical results (i.e., "our rollout is better than whatever inspired our rollout", or "look at our rollout, it outperforms popular methods"). This work is a solid generalization of popular rollout papers, at least the ones I'm familiar with. Thus I think that this paper is a strong contribution to the field, although its accessibility could be improved (with intuition, examples, etc.). The potential software library as well as theoretical framework provides a basis for further exploration in terms of both DNNs, and "traditional" iterative algorithms, which can be viewed as infinitely long rollouts. I would look forward to seeing this paper presented.

Reviewer 2



Summary: The authors provide theoretical and empirical analysis of a new rollout scheme for running inference on arbitrary neural network graphs. Additionally, they develop a general theory of rollouts for networks graphs, and (will) provide an open source toolbox implementing their method. Strengths: The paper is exceptionally clear and well-supported by numerical experiments. The authors even include a snapshot of the eventual open source tool that they will provide to use the method. Weaknesses: For someone unfamiliar with the general art (or witchcraft) of unrolling networks, the choice of baselines make sense, but seem…sparse? Does it make sense to, for example, additionally perform a random, representative hybrid roll-out to compare against the fully-sequential and fully-streaming data in Fig. 3? More specifically, it’s unclear to me how the gains scale with amount-of-streaming, besides the obvious fact that more streaming = better. This is a comparatively minor point though! Overall, the paper is an excellent contribution to the community, and I look forward to seeing the open source package!

Reviewer 3



Update after rebuttal: I appreciate that the authors addressed my concerns in their rebuttal. I hope that the authors can clarify the exposition + add examples as discussed, which would make the paper significantly more accessible. --- Summary: This paper analyzes different ways to unroll recurrent neural networks with skip connections and self-loops. A main motivation is to increase the efficiency (e.g., response time) of the network during training/inference. A rollout is a graph that captures the functional dependency of network nodes over time. The authors argue that there are different possible rollouts that have different quality (e.g., response time), introduce mathematical definitions to describe rollouts (e.g., validity / model-parallelizable) and analyze rollouts theoretically and experimentally. In my understanding, the actual conclusion of the paper seems to be: the streaming ("R \equiv 1") is the best, e.g., Theorem in L192 states that the streaming rollout achieves the lowest response time over the entire graph. The experiments seem to support that conclusion. Note that how to obtain the streaming rollout is not clearly stated by the authors, although the Thm in L192 seems to suggest a working rule for obtaining it. Pro: Originality/Significance: - I'm not aware of earlier work that analyzes this low-level implementation issue, but it is worthwhile to analyze this for optimization purposes. - Overall, the paper is grammatically well written. Con: Quality: - The writing is a bit dense in places and the payoff from introducing all the different mathematical definitions seems to be the Thm in L192. Clarity: - It's not so clearly explained what the computational overhead is of the streaming rollout R \equiv 1: is there any since we are using more edges from the functional graph? - The paper lacks some simple examples to motivate / illustrate the introduced mathematical definitions. E.g., what does R \equiv 1 represent / what is the intuition for (1)? A simple example would be useful. The same for "tableau" and "state": a simple example to elucidate the definitions would be helpful. - Why is the sequential ("R \equiv 0") rollout as in the top row in Fig 1? Maybe a detailed discussion / description here would be helpful. L212: "single images are already sufficient for almost perfect classification results" --> I don't know what is meant here. - Theorems are not numbered. - Some formulas are not numbered. Reproducibility: - How is this implemented in framework XYZ? - The authors don't seem to discuss how operationally to obtain the streaming rollout automatically from the functional definition of a computation graph. How do you do that? - It does not seem straightforward to re-implement this without error. Typos: - L122: express -> expressed